# Nigeria healthcare worker SARS-CoV-2 serology study: Results from a prospective, longitudinal cohort

**Dike B. Ojji**[1,2]\*, **Amelia Sancilio**[3], **Gabriel L. Shedul**[1], **Ikechukwu A. Orji**[1],
**Aashima Chopra**[4], **Joel Abu**[1], **Blessing Akor**[1], **Nana Ripiye**[1], **Funmi Akinlade**[5],
**Douglas Okoye**[1], **Emmanuel Okpetu**[1], **Helen Eze**[1], **Emmanuel Odoh**[6], **Abigail
S. Baldridge**[4,7], **Priya Tripathi**[4], **Haruna Abubakar**[2], **Abubakar M. Jamda**[8], **Lisa
R. Hirschhorn**[9], **Thomas McDade**[3], **Mark D. Huffman**[4,10,11]

1 Department of Internal Medicine, Faculty of Clinical Sciences, Federal Capital Territory, University of Abuja, Abuja, Nigeria, 2 Cardiovascular Research Unit, University of Abuja and University of Abuja Teaching Hospital, Federal Capital Territory, Abuja, Nigeria, 3 Northwestern University Department of Anthropology and Institute for Policy Research, Evanston, Illinois, United States of America, 4 Northwestern University Department of Preventive Medicine, Chicago, Illinois, United States of America, 5 Johns Hopkins University, Baltimore, Maryland, United States of America, 6 Abaji Department of Primary Care, Abaji, Federal Capital Territory, Abuja, Nigeria, 7 Bluhm Cardiovascular Institute Clinical Trials Unit, Northwestern Medicine, Chicago, Illinois, United States of America, 8 Department of Community Medicine, Faculty of Clinical Sciences, Federal Capital Territory, University of Abuja, Abuja, Nigeria, 9 Northwestern University Department of Medical Social Sciences, Chicago, Illinois, United States of America, 10 Northwestern University Department of Medicine, Chicago, Illinois, United States of America, 11 The George Institute for Global Health, University of New South Wales, Sydney, Australia

\* dike.ojji@uniabuja.edu.ng

**Data Availability Statement:** Individual participant data on the results reported in this article are available at OSF: https://osf.io/dwfn8 and the identifier: DOI 10.17605/OSF.IO/DWFN8.

## Abstract

Healthcare workers, both globally and in Nigeria, have an increased risk for SARS-CoV-2 infection compared with the general population due to higher risk contacts, including occupational exposures. In addition, primary healthcare workers represent an important group for estimating prior infection to SARS-CoV-2 because they work at the first point-of-contact for most patients yet have not been included in prior COVID-19 seroepidemiology research in Nigeria. We sought to evaluate baseline seroprevalence, rates of seroconversion (IgG- to IgG+) and seroreversion (IgG+ to IgG-), change in IgG concentration at 3- and 6-month follow-up, and factors associated with seropositivity. From June 2020 to December 2020, we conducted a longitudinal seroepidemiology study among frontline health care workers in Nigeria using a validated dried blood spot assay. Among 525 participants, mean (SD) age was 39.1 (9.7) years, 61.0% were female, and 45.1% were community health workers. The six-month follow-up rate was 93.5%. Seropositivity rates increased from 31% (95% CI: 27%, 35%) at baseline to 45% (95% CI: 40%, 49%) at 3-month follow-up, and 70% (95% CI: 66%, 74%) at 6-month follow-up. There was a corresponding increase in IgG levels from baseline (median = 0.18 ug/mL) to 3-month (median = 0.35 ug/mL) and 6-month follow-up (median = 0.59 ug/mL, $P_{trend}$ < .0001). A minority of participants reported symptoms from February 2020 until baseline (12.2%) or during 3-month (6.6%) or 6-month (7.5%) follow-up. only 1 participant was hospitalized. This study demonstrated high baseline, 3-month and 6-month follow-up prevalence of IgG antibodies to SARS-CoV-2 during the first two waves of

**Funding:** This study was sponsored by the National Heart, Lung, and Blood Institute (R01HL144708) with additional support from the Robert J Havey MD Institute for Global Health and Northwestern University Clinical and Translational Science Institute (UL1TR00422). The funders had no role in the design, conduct of the study; collection, management, analysis, and interpretation of the data; preparation, review, or approval of the manuscript; or decision to submit the manuscript for publication.

**Competing interests:** I have read the journal's policy and one of the authors, MDH has the following competing interests:" has patents pending for combination therapy for the treatment of heart failure (HFpEF Poly diuretic) with application serial number: 63/261,121. The George Institute for Global Health has a patent, license, and has received investment funding with intent to commercialize fixed-dose combination therapy through its social enterprise business, George Medicines. The other authors do not report any disclosures.

the COVID-19 pandemic in Nigeria among a cohort of unvaccinated frontline healthcare workers, including primary healthcare workers despite low symptomatology. These results may have implications in state- and national-level disease pandemic modeling.

**Trial registration:** NCT04158154.

## Introduction

As of April 1, 2022, there were >489 million cases of COVID-19 diagnosed worldwide [1]. After Nigeria's index case was announced on February 28, 2020 [2], there have been 255,415 confirmed cases of COVID-19 through April 1, 2022. However, the true number of SARS-CoV-2 infections in Nigeria is unknown and may be nearly 1,000-fold higher than the number of confirmed cases [3, 4]. Seroepidemiology studies can estimate the prevalence of prior infections to evaluate the burden of disease and to inform disease modeling and mitigation efforts. A household seroprevalence study led by the Nigeria Center for Disease Control and Prevention in 4 states in Nigeria (Gombe, Enugu, Nasarawa and Lagos) demonstrated prior infection rates based on the presence of IgG antibodies ranging from 9% in Gombe to 23% in Lagos as of October 2020 [5]. National seroprevalence has been estimated to be 78.9% (95% CI: 77.7, 80.0) based on data collected in Q3 2021 [6].

Healthcare workers, both globally and in Nigeria, have an increased risk for SARS-CoV-2 infection compared with the general population due to higher risk contacts, including occupational exposures. Epidemiological research on COVID-19 among healthcare workers in Nigeria has demonstrated rates of current SARS-CoV-2 infection as high as 21.7% among healthcare workers in COVID-19 isolation and treatment centers from January 2020 to August 2020 in Nasarawa State using reverse transcriptase polymerase chain reaction testing [7]. In April 2020, one study among tertiary center healthcare workers in Ibadan showed a seroprevalence of IgG antibodies to SARS-CoV-2 of 45.1% [8]. It is uncertain how these rates compare with primary care healthcare workers, who represent an important group for estimating prior infection to SARS-CoV-2 because they work at the first point-of-contact for most patients, especially those who are either asymptomatic or minimally symptomatic. Prior seroprevalence research in Nigeria has also more frequently relied upon samples collected through venipuncture by trained research or clinical staff, which limits the reach of research and does not represent outlying areas with limited resources. Collection of blood samples using a fingerstick technique has been demonstrated to be feasible and effective in estimating community-level IgG seropositivity and levels of antibodies to SARS-CoV-2 [9]. We sought to use this method for estimating IgG seroprevalence to SARS-CoV-2 among frontline healthcare workers who were part of an existing research program in the Federal Capital Territory of Nigeria [10].

The objectives of this study were to create a longitudinal cohort of Nigerian frontline healthcare workers at the primary and tertiary care levels in the first year of the COVID-19 pandemic to: 1) describe the baseline prevalence of anti-SARS2 IgG serology using a dried blood spot-based assay, overall and by sociodemographic, 2) quantify COVID-19 symptoms, diagnostics, and clinical outcomes, 3) assess rates of seroconversion (IgG- to IgG+) and sereoreversion (IgG+ to IgG-) and change in IgG concentrations at 3- and 6-month follow-up, and 4) identify factors associated with seropositivity.

## Materials and methods

### Study design

We conducted a prospective cohort study to evaluate the prevalence of SARS-CoV-2 IgG antibodies to the receptor binding protein domain of the SARS-CoV-2 spike protein using dried

blood spot samples that were collected from health care workers in the Federal Capital Territory of Nigeria at baseline, 3-month, and 6-month follow-up. Health care workers were recruited from primary and tertiary care facilities participating in and affiliated with an ongoing research study (NCT04158154). The protocol was adapted from the World Health Organization population-based and health care worker seroepidemiology protocols. The clinical and data coordinating center was based at the University of Abuja Teaching Hospital, which provided ethical review and approval of the study procedures, along with Federal Capital Health Research Ethics Committee (FHREC/2020/01/58/30-06-20) and the Institutional Review Board of Northwestern University (STU00212900).

## Study population

From July 2020 to December 2020, health care workers from 60 primary health care centers and University of Abuja Teaching Hospital were identified through the study team's existing research infrastructure. The approach to the site-level sampling frame among primary health care centers, which was conducted in 2019 as part of an ongoing research study, has been published [10]. Briefly, among the 243 primary health care centers, the study performed multi-stage probability sampling across the 6 area councils and 62 wards in the Federal Capital Territory. In Nigeria, most primary health care centers are publicly funded institutions that provide frontline care. Academic centers like the University of Abuja Teaching Hospital function as referral centers. Due to the opportunity for exposure to SARS-CoV-2, health care workers were defined as all health care professionals and auxiliary workers. Health care workers who had potential patient contact at any time before study enrollment were eligible and include physicians, nurses, community health officers, community health extension workers, record officers, laboratory technicians, pharmacists, pharmacy technicians, and other allied workers (e.g., environmental services, security). In primary healthcare centers, nurses and community health extension workers provide most frontline clinical care with oversight from physicians and community health officers and support from other staff.

Potential participants received an email or text message invitation to participate in this prospective cohort study. Eligibility criteria included: 1) full-time employment as a health care worker to minimize losses to follow-up and 2) willingness and ability to provide informed consent. We conducted a convenience sample based on availability for testing and willingness to participate but sought to recruit a diverse sample based on age, sex, and health care worker type (e.g., physician or advanced practice provider, nurse or community health worker, and auxiliary staff). We targeted a sample size of 525 participants because this would produce a 95% confidence interval with a width of 0.039 if the baseline prevalence of anti-SARS-CoV-2 IgG serology were 0.05.

The study was reviewed and approved by the University of Abuja Teaching Hospital and Northwestern University Institutional Review Boards. REDCap was used to obtain informed consent and questionnaire completion using electronic tablets that were already based at the recruitment sites as part of an existing research program. This system also facilitated reporting results back to participants, which was supplemented by telephone calls to participants who did not have email addresses. Trained study staff conducted interviewer-administered surveys after informed consent was received through an electronic signature. Variables included socio-demographic factors, such as age, sex, living situation, medical history, potential occupational and personal exposures, SARS-CoV-2 testing history, and presence or recent history of symptoms which could be associated with SARS-CoV-2 infection [9]. Surveys documenting exposures, symptoms, history of COVID-19 diagnostic testing, and clinical outcomes were repeated at 3-month and 6-month follow-up.

## Serological testing

Blood samples were collected by trained study staff who adhered to infection prevention and control procedures during sample collection. These procedures included appropriate hand hygiene, correct use of medical or respiratory face masks or shields, use of gloves, and best phlebotomy practices. The fingerstick method was used to capture 5 dried blood spot samples from each participant, which was collected using Whatman #903 filter paper for storage. Fingerstick dried blood spot samples were collected at baseline, 3-month, and 6-month follow-up, and shipped to a central laboratory in a single batch at each time point for sample analysis. Standard operating procedures for a validated dried blood spot assay specific to SARS-CoV-2 were used according to a published protocol [9].

Samples were analyzed using a validated dried blood spot assay specific to SARS-CoV-2 [9]. This method was adapted from a quantitative ELISA that was granted Emergency Use Authorization from the US Food and Drug Administration to measure SARS-CoV-2 antibodies in serum and that was determined to not cross-react with other common coronavirus strains [12]. Briefly, the ELISA method was validated using an antibody (CR3022) with defined affinity to the receptor binding domain of the spike protein [9, 11, 12]. To determine seropositivity for the presence of SARS-CoV-2 antibodies, the cut-off was set at the optical density value produced by the 0.39 μg/ml CR3022 calibrator. This low seropositive cut-off is greater than 3 standard deviations above optical density values for 23 known negative samples collected in 2018, and it is well above the assay's lower limit of detection. The method shows strong agreement between matched dried blood spot and serum samples from 17 individuals ($R^2 = 0.9945$).

## Statistical analysis

Participants' characteristics are reported as means (standard deviation) for continuous variables and as proportions (95% confidence intervals [CI]) for categorical variables, overall and by sex. The prevalence of SARS-CoV-2 seropositivity was estimated as the proportion of participants with positive results based on the previously published threshold of 0.40 μg/m [9]. Trend tests for evaluating changes in median IgG levels over time were performed to compare IgG levels overall and by baseline seropositivity status. Seropositivity rates were compared by sociodemographic characteristics, home and work council area, and occupational and non-occupational exposures to SARS-CoV-2. Seroconversion (IgG- to IgG+) and seroreversion (IgG+ to IgG-) were also reported for all the participants for all three time points.

Unadjusted and adjusted multivariate logistic regression models were created to evaluate association between baseline characteristics and IgG seropositivity to SARS-CoV-2 at baseline and each follow-up. Sociodemographic characteristics, including area council of participants' work (work area council), and medical history, including COVID-19 testing and diagnosis, were assessed as risk factors associated with SARS-CoV-2 seropositivity. Multivariate models were adjusted for age, sex, BMI, occupation, work area council, and history of immunocompromised state based on a previous report [9, 11], as well as adjustment for within within-work location (i.e., primary health care center or hospital) clustering. Participants' home area council was considered in these models but was collinear with work council area. Mixed-effect multivariable logistic regression models were also created to evaluate associations between participant characteristics, including potential occupational exposures, and IgG seropositivity of SARS-CoV-2 to account for repeated SARS-CoV-2 serology tests among participants at baseline, 3-month, and 6-month follow-up.

A complete case analysis was performed for the primary analysis. Sensitivity analyses were performed to evaluate the robustness of the results using all available data, including participants with any missing serology information. A two-sided p value <0.05 was used to define

statistical significance. All analyses were conducted using SAS version 9.4 (SAS, Cary, NC, USA) and R version 4.0.5 (R Foundation for Statistical Computing, Vienna, Austria).

## Results

The flowchart of participants is reported in S1 Fig, which demonstrates 93.5% follow-up at 6 months among 525 baseline participants. Table 1 describes participants' baseline characteristics overall and stratified by sex. Mean (SD) age was 39.1 (9.7) years, 61.0% of participants were female, and 45.1% were community health workers. Females had a higher mean body mass index (28.2 [6.4] kg/m$^2$ versus 25.7 [4.7] kg/m$^2$, p<0.01), were more likely to be a community health worker (47.2% versus 42.0%, p<0.01), and were less likely to have had postgraduate education (11.6% versus 22.4%, p = 0.01) when compared with males. Overall, the proportions of participants with a self-reported history of chronic health conditions were relatively low, including hypertension (14.1%), diabetes mellitus (4.2%), and immunocompromised state (1.9%), which were similar between sexes. Health care workers were more likely to live (32.0%) or work (34.1%) in Gwagwalada council area compared with other council areas (p<0.01 for both), which was due to pre-specified, additional sampling at the University of Abuja Teaching Hospital (58/525, or 11% of participants from that council area). At baseline, 12.6% of participants reported any preceding symptoms (S1 Table), which were most commonly fever, body aches, or weakness. There was with no difference in the prevalence of symptoms among individuals above and below the median age (40 years).

Table 2 reports the proportions (95% CI) of participants who had IgG seropositivity to SARS-CoV-2 at baseline, 3-month, and 6-month follow-up by participants' work and home area councils. Seropositivity rates increased from 31% (95% CI: 27%, 35%) at baseline to 45% (95% CI: 40%, 49%) at 3-month follow-up, and 70% (95% CI: 66%, 74%) at 6-month follow-up. The lowest proportions were in Kuje work council area for baseline (22%), 3-month (36%), and 6-month follow-up (58%). The highest proportion was in Abuja Municipal Area Council at 6-month follow-up (82%). The patterns of seropositivity trends and proportions were similar based on participants' home area council.

Fig 1 demonstrates seroconversion and seroreversion of IgG to SARS-CoV-2 at baseline, 3-month, and 6-month follow-up. Seroconversion (IgG- to IgG+) was observed in 45.6% of participants in this cohort and was more common than seroreversion (IgG+ to IgG-; 5.2%). Fig 2 shows the increase in IgG levels at baseline (median = 0.18 ug/mL), 3-month (median = 0.35 ug/mL), and 6-month follow-up (median = 0.59 ug/mL) among the overall study sample ($P_{trend}$ < .0001). S2 and S3 Figs show the trends in IgG levels among individuals who were seropositive and seronegative at baseline, respectively. Median (IQR) IgG levels among individuals who were seropositive at baseline were similar at baseline (0.87 [0.59, 1.45] ug/mL) and 6-month follow-up (0.79 [0.45, 1.51] ug/mL). Among individuals who were seronegative at baseline, median (IQR) IgG levels increased from 0.09 (0.07, 0.20) ug/mL at baseline to 0.55 (0.25, 0.94) ug/mL at 6-month follow-up.

S2 Table shows baseline, 3-month, and 6-month follow-up data on COVID-19 symptoms, diagnostic testing, diagnosis, and clinical outcomes. The proportion of individuals with symptoms consistent with COVID-19 was 12.2% at baseline, 6.6% at 3-month follow-up, and 7.5% at 6-month follow-up. Diagnostic testing and diagnosis of COVID-19 were uncommon, and only 1 participant was hospitalized.

The unadjusted and adjusted logistic regression model results to evaluate factors associated with seropositivity at baseline and at any time point are reported in Table 3. There were no associations observed between sociodemographic, occupation, or home or work council and seropositivity status, though the confidence intervals were wide for some factors.

**Table 1. Baseline study participants' characteristics, overall and by sex.**

| Baseline Characteristic | Total | Female | Male | P-value |
|---|---|---|---|---|
| | (N = 525) | (N = 320) | (N = 205) | |
| Age, mean (SD), years | 39.1 (9.7) | 39.1 (10.0) | 39.2 (9.6) | 0.93 |
| Body mass index, mean (SD), kg/m$^2$ | 27.2 (5.9) | 28.2 (6.4) | 25.7 (4.7) | < .01 |
| Married, No. (%) | 392 (74.6) | 238 (74.4) | 154 (75.1) | 0.84 |
| Monthly household income >39,000 Naira, No. (%) | 343 (65.3) | 203 (63.4) | 140 (68.3) | 0.25 |
| Insurance, No. (%) | | | | 0.01 |
| Government | 139 (26.4) | 73 (22.8) | 66 (32.2) | |
| Unsure/None | 377 (71.8) | 244 (76.3) | 133 (64.9) | |
| Private | 9 (1.7) | 3 (0.9) | 6 (2.9) | |
| Occupation, No. (%) | | | | < .01 |
| Community health worker | 237 (45.1) | 151 (47.2) | 86 (42.0) | |
| Physician or Nurse | 109 (20.7) | 75 (23.4) | 34 (16.6) | |
| Other* | 179 (34.1) | 94 (29.4) | 85 (41.5) | |
| Education, No. (%) | | | | 0.01 |
| Diploma and below | 345 (65.7) | 221(69.0) | 124 (60.5) | |
| Under graduation | 97 (18.5) | 62 (19.4) | 35 (17.1) | |
| Post-graduation | 83 (15.8) | 37 (11.6) | 46 (22.4) | |
| History of hypertension, No. (%) | 74 (14.1) | 48 (15.0) | 26 (12.7) | 0.46 |
| History of diabetes mellitus, No. (%) | 22 (4.2) | 17 (5.3) | 5 (2.4) | 0.11 |
| History of obesity, No. (%) | 46 (8.7) | 41 (12.8) | 5 (2.4) | < .01 |
| Asthma medication use, No. (%) | 13 (2.5) | 9 (2.8) | 4 (2.2) | 0.52 |
| History of immunocompromised state, No. (%) | 10 (1.9) | 8 (2.5) | 2 (1.1) | 0.21 |
| History of liver disease, No. (%) | 1 (0.2) | 1 (0.3) | 0 | Not estimable |
| History of cancer, No. (%) | 1 (0.2) | 1 (0.3) | 0 | Not estimable |
| History of chronic kidney disease, No. (%) | 1 (0.2) | 0 | 1(0.5) | Not estimable |
| History of heart failure, No. (%) | 1 (0.2) | 1 (0.3) | 0 | Not estimable |
| History of coronary heart disease, No. (%) | 1 (0.2) | 1 (0.3) | 0 | Not estimable |
| Home council area, No. (%) | | | | < .01 |
| Abaji | 53 (10.1) | 29 (9.1) | 24 (11.7) | |
| Abuja Municipal Area Council | 111 (21.1) | 82 (25.6) | 29 (14.1) | |
| Bwari | 62 (11.8) | 42 (13.1) | 20 (9.8) | |
| Gwagwalada | 168 (32.0) | 92 (28.8) | 76 (37.1) | |
| Kuje | 59 (11.2) | 39 (12.2) | 20 (9.8) | |
| Kwali | 56 (10.6) | 27 (8.4) | 29 (14.1) | |
| Other | 16 (3.0) | 9 (2.8) | 7 (3.4) | |
| Work Council Area, No. (%) | | | | < .01 |
| Abaji | 56 (10.6) | 31 (9.7) | 25 (12.2) | |
| Abuja Municipal Area Council | 120 (22.8) | 87 (27.2) | 33 (16.1) | |
| Bwari | 56 (10.6) | 39 (12.2) | 17 (8.3) | |
| Gwagwalada** | 179 (34.1) | 93 (29.1) | 86 (42.0) | |
| Kuje | 56 (10.6) | 38 (11.9) | 18 (8.8) | |
| Kwali | 58 (11.0) | 32 (10.0) | 26 (12.7) | |

*Including pharmacist, laboratory technician, non-physician clinician/paramedical professionals, among other allied health staff.

**58 (34.5%) from the University of Abuja Teaching Hospital, 110 (63.5%) from primary healthcare centers.

**Table 2. Proportion (95% CI) of IgG seropositivity to SARS-CoV-2 at baseline, 3-month, and 6- month follow-up by participants' work and home council areas.**

| Council Area (Work) | Baseline | Month 3 | Month 6 |
|---|---|---|---|
| | (n = 490) | (n = 493) | (n = 489) |
| Abaji | 0.39 (0.26–0.52) | 0.49 (0.36–0.63) | 0.73 (0.62–0.85) |
| Abuja Municipal Area Council | 0.25 (0.17–0.34) | 0.46 (0.36–0.55) | 0.82 (0.75–0.89) |
| Bwari | 0.32 (0.20–0.44) | 0.48 (0.35–0.61) | 0.63 (0.50–0.80) |
| Gwagwalada | 0.31 (0.24–0.38) | 0.44 (0.36–0.52) | 0.67 (0.59–0.75) |
| Kuje | 0.22 (0.11–0.33) | 0.36 (0.24–0.49) | 0.58 (0.45–0.71) |
| Kwali | 0.38 (0.25–0.50) | 0.44 (0.32–0.57) | 0.62 (0.49–0.75) |
| **Overall** | **0.31 (0.27–0.35)** | **0.45 (0.40–0.49)** | **0.70 (0.66–0.74)** |
| **Council Area (Home)** | | | |
| Abaji | 0.42 (0.28–0.55) | 0.53 (0.39–0.66) | 0.73 (0.60–0.85) |
| Abuja Municipal Area Council | 0.23 (0.14–0.31) | 0.48 (0.38–0.57) | 0.82 (0.75–0.89) |
| Bwari | 0.35 (0.23–0.47) | 0.45 (0.33–0.57) | 0.65 (0.53–0.77) |
| Gwagwalada | 0.30 (0.23–0.37) | 0.43 (0.35–0.51) | 0.68 (0.61–0.76) |
| Kuje | 0.28 (0.16–0.39) | 0.35 (0.23–0.47) | 0.56 (0.43–0.69) |
| Kwali | 0.36 (0.24–0.49) | 0.47 (0.34–0.60) | 0.67 (0.55–0.79) |
| Other | 0.27 (0.04–0.49) | 0.33 (0.10–0.57) | 0.73 (0.50–0.96) |
| **Overall** | **0.31 (0.27–0.35)** | **0.45 (0.40–0.49)** | **0.70 (0.66–0.74)** |

*Mishandled, missing, or insufficient samples were not included in this analysis.

Results from the mixed effects models were generally similar to the multivariable logistic regression models (S4 Fig), except for a lower odd of seropositivity among individuals who worked in Kuje compared with Abaji area council (OR = 0.50, 95%CI: 0.29–0.88). Based on potential occupational exposures, no factors were significantly associated with seropositivity, though the confidence intervals were wide for some factors (S5 Fig).

## Discussion

This report demonstrates a high baseline prevalence (31%) of IgG antibodies to SARS-CoV-2 among a cohort of frontline healthcare workers in the Federal Capital Territory early in the COVID-19 pandemic in Nigeria using a dried blood spot-based assay. Seroprevalence substantially increased over the 6-month follow-up period (70%), a period that included the second wave of cases and preceded the start of COVID-19 vaccinations in Nigeria (March 5, 2021) [13]. While genomic surveillance was limited in Nigeria during the study period, results from phylogenetic analyses suggest that variants B.1.1.7 (Alpha) and B.1.525 (Eta) were the most commonly circulating variants introduced in fall 2020 [14].

There were geographic differences in the 6-month seroprevalence rates across work, or occupation, area councils within the Federal Capital Territory, which were highest in Abuja Municipal Area Council (82%) and lowest in Kuje (58%). The high seroprevalence rate observed in Abuja Municipal Area Council may be due, at least in part, to the higher proportion of participants living outside of their work area council compared with other area councils. This finding suggests that higher mobility from work-related transportation was associated with increased exposure to SARS-CoV-2. Other sociodemographic or occupational characteristics were not associated with seropositivity in this study, which was conducted before access to vaccines and highlights the need for both personal protective equipment and rapid testing access for all healthcare workers.

A minority (12.5%) of participants had symptoms consistent with COVID-19 infection from February 2020 to the baseline period (July 2020) with similar rates among individuals

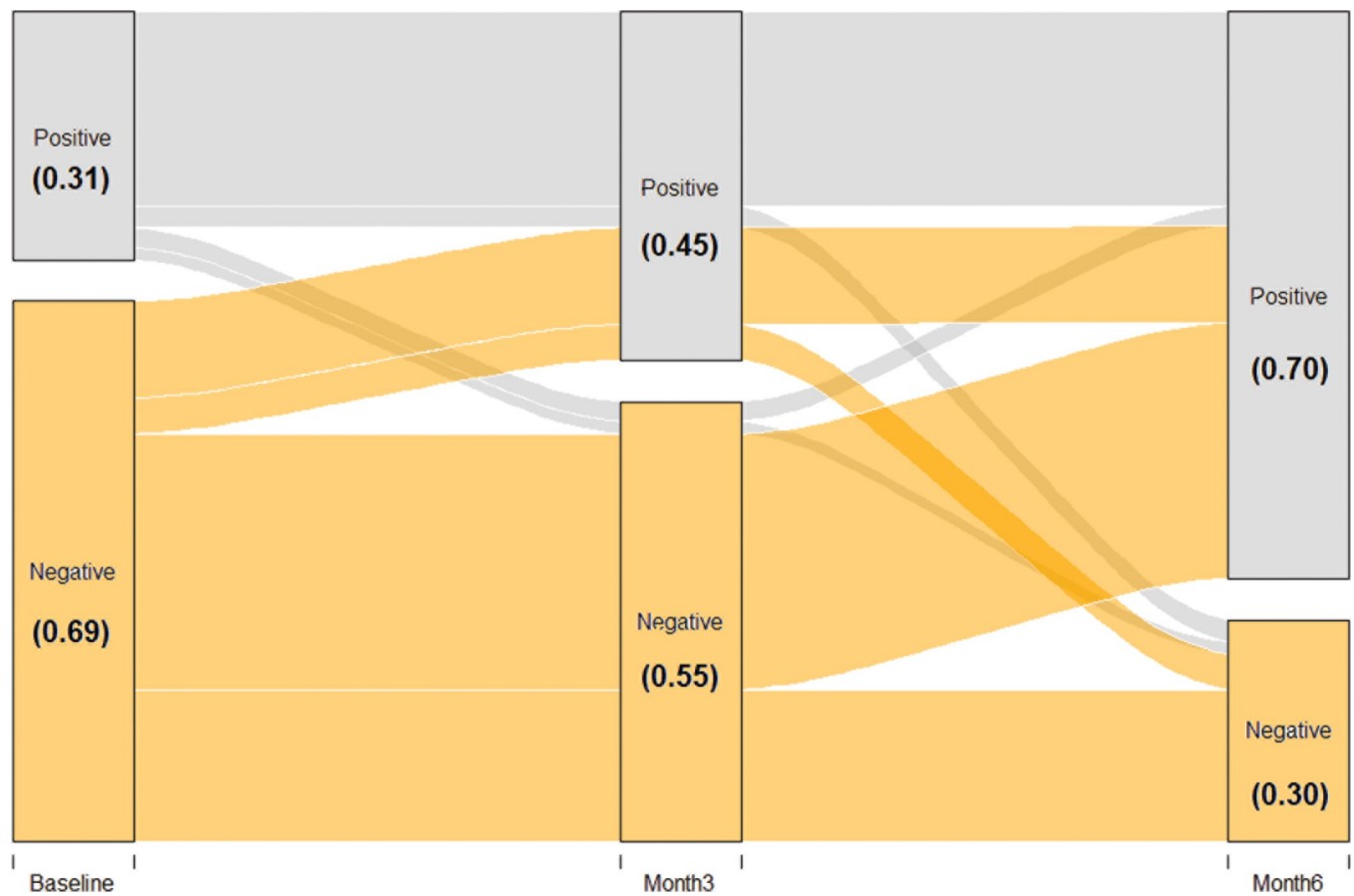

**Fig 1. Bidirectional seroconversion of IgG to SARS-CoV-2 at baseline, 3-month, and 6-month follow-up based on complete case analysis.**

who were seropositive and seronegative at baseline. Only one participant was hospitalized, and there were no deaths in this cohort over the study period. The reasons for the low rate of symptoms are uncertain but may include the relatively young mean age of the cohort (39.1 years old), low proportion of comorbid conditions associated with worse clinical outcomes from COVID-19, infections with strains of SARS CoV-2 that cause less severe disease or recall bias [3]. The low symptom burden in this population warrants further investigation. Among individuals who were seropositive at baseline, there was a modest decline in IgG levels over time, though this did not reach statistical significance.

Most participants were based in primary healthcare settings where asymptomatic or minimally symptomatic patients may have sought and received care from members of this cohort over the study period. Throughout the pandemic, there has been limited personal protective equipment available in Nigeria [15], an issue that was greatest early in the pandemic and which further exacerbated exposure risks. Identification, isolation, and treatment of healthcare workers who have COVID-19 is also important to ensure a sufficient healthcare workforce to respond to the pandemic, as well as to minimize nosocomial and community spread.

Global weighted averages from systematic reviews of seroprevalence studies estimate 8% (95% CI: 6–10%) of healthcare workers (through August 2020, prior to routine vaccination) [16], had IgG antibodies to SARS-Co-V-2 compared with a weighted median of 4.5% (interquartile range: 2.4–8.4%) of the general population (through December 2020) [17]. A 2021 scoping review of seroprevalence studies among healthcare workers in 11 African countries,

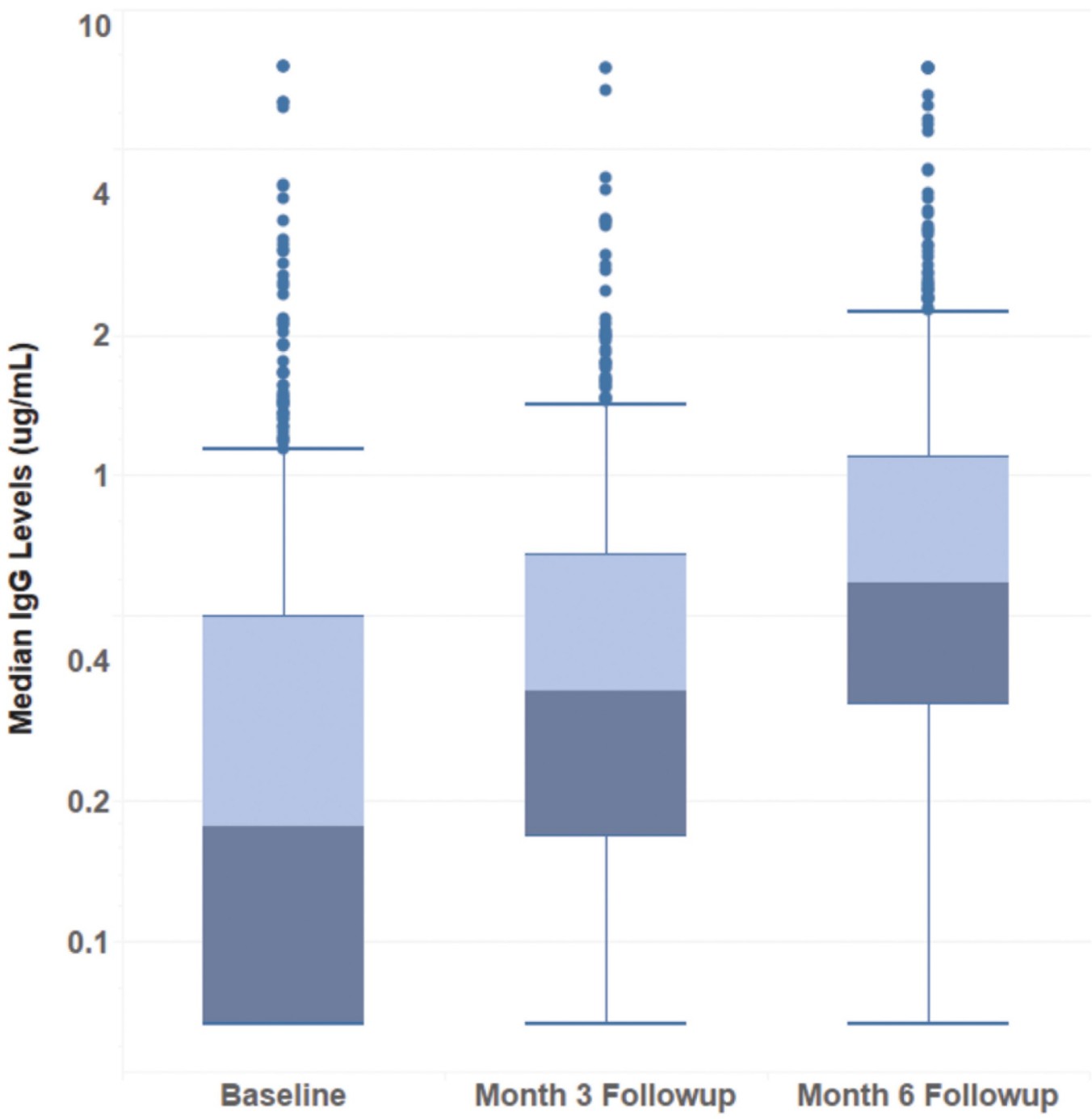

**Fig 2. Distribution of IgG levels at baseline (median = 0.178 ug/mL), month 3 (median = 0.348 ug/mL), and month 6 follow-up (median = 0.591 ug/mL) based on complete case analysis.**

which overlapped with the current study period (April 2020 to December 2020), demonstrated a seroprevalence range of 0% to 45.1% [18]. The highest estimate was conducted in April 2020 in a Nigerian tertiary care facility (i.e., University College Hospital in Ibadan) and showed a higher prevalence than the baseline result of the current study (45.1% [95% CI: 36.5%, 54.0%] versus 31% [95% CI: 27%, 35%]) using an ELISA technique for qualitative SARS-CoV-2 spike

**Table 3. Multivariate logistic regression model to evaluate association between baseline characteristics and IgG seropositivity to SARS-CoV-2 at baseline and at any time point.**

| Variables | Seropositive at baseline (n = 150) | | | Seropositive at any time point (n = 402) | | |
|---|---|---|---|---|---|---|
| | Model 1 | Model 2 | Model 3 | Model 1 | Model 2 | Model 3 |
| | Unadjusted OR (95%CI) | Adjusted* OR (95%CI) | Adjusted† OR (95%CI) | Unadjusted OR (95%CI) | Adjusted* OR (95%CI) | Adjusted† OR (95%CI) |
| AGE, BY YEAR | 1.004 (0.98–1.02) | 1.004 (0.98–1.02) | 1.01 (0.99–1.03) | 0.99 (0.97–1.02) | 0.99 (0.97–1.02) | 1.00 (0.98–1.02) |
| SEX | | | | | | |
| MALE | Reference | Reference | Reference | Reference | Reference | Reference |
| FEMALE | 0.93 (0.63–1.38) | 0.93 (0.63–1.38) | 1.02 (0.67–1.57) | 0.98 (0.63–1.52) | 0.97 (0.62–1.52) | 1.02 (0.64–1.66) |
| BMI, PER 1 KG/M$^2$ | 0.98 (0.95–1.02) | 0.98 (0.95–1.01) | 0.98 (0.95–1.02) | 1.00 (0.96–1.04) | 1.00 (0.96–1.04) | 0.99 (0.96–1.04) |
| MARITAL STATUS | | | | | | |
| MARRIED | Reference | Reference | Reference | Reference | Reference | Reference |
| NOT MARRIED | 0.84 (0.55–1.30) | 1.32 (0.81–2.15) | 1.47 (0.88–2.48) | 0.82 (0.50–1.35) | 1.21 (0.65–1.84) | 1.42 (0.78–2.60) |
| HOUSEHOLD INCOME | | | | | | |
| <39,000 NAIRA | Reference | Reference | Reference | Reference | Reference | Reference |
| ≥39,000 NAIRA | 1.06 (0.71–1.60) | 1.02 (0.64–1.63) | 0.93 (0.56–1.55) | 1.02 (0.65–1.60) | 1.10 (0.65–1.84) | 1.09 (0.62–1.95) |
| INSURANCE, NO. (%) | | | | | | |
| GOVERNMENT | Reference | Reference | Reference | Reference | Reference | Reference |
| UNSURE/NONE | 1.24 (0.79–1.93) | 1.33 (0.83–2.13) | 1.65 (0.95–2.85) | 1.02 (0.63–1.66) | 0.99 (0.59–1.65) | 1.15 (0.63–2.13) |
| PRIVATE | 0.33 (0.04–2.70) | 0.34 (0.04–2.84) | 0.40 (0.04–3.48) | 2.30 (0.28–19.13) | 2.20 (0.26–18.52) | 2.95 (0.32–27.08) |
| OCCUPATION | | | | | | |
| COMMUNITY HEALTH WORKER | Reference | Reference | Reference | Reference | Reference | Reference |
| DOCTOR/NURSE | 1.19 (0.72–1.97) | 1.19 (0.71–2.01) | 1.35 (0.77–2.39) | 0.70 (0.40–1.22) | 0.71 (0.40–1.26) | 0.79 (0.38–1.36) |
| OTHER** | 0.87 (0.56–1.36) | 0.86 (0.55–1.35) | 0.97 (0.60–1.54) | 0.83 (0.51–1.35) | 0.82 (0.50–1.35) | 0.88 (0.47–1.33) |
| EDUCATION | | | | | | |
| DIPLOMA AND BELOW | Reference | Reference | Reference | Reference | Reference | Reference |
| UNDER GRADUATION | 1.12 (0.68–1.84) | 1.10 (0.66–1.81) | 1.03 (0.60–1.78) | 1.06 (0.59–1.90) | 1.07 (0.60–1.93) | 1.06 (0.55–2.05) |
| POST-GRADUATION | 0.80 (0.46–1.40) | 0.75 (0.42–1.35) | 0.70 (0.36–1.35) | 0.70 (0.39–1.23) | 0.70 (0.38–1.27) | 0.55 (0.27–1.09) |
| COUNCIL AREA (WORK) | | | | | | |
| ABAJI | Reference | Reference | Reference | Reference | Reference | Reference |
| ABUJA MUNICIPAL AREA COUNCIL | 0.53 (0.27–1.05) | 0.53 (0.27–1.07) | 2.72 (0.20–36.32) | 1.34 (0.57–3.09) | 1.34 (0.57–3.11) | 0.98 (0.06–15.83) |
| BWARI | 0.73 (0.34–1.59) | 0.74 (0.34–1.61) | 1.64 (0.11–25.5) | 0.81 (0.33–1.99) | 0.81 (0.33–2.01) | 0.53 (0.03–9.47) |
| GWAGWALADA | 0.69 (0.37–1.31) | 0.69 (0.36–1.30) | 2.81 (0.25–31.74) | 0.90 (0.42–1.93) | 0.91 (0.2–1.96) | 1.72 (0.14–20.45) |
| KUJE | 0.43 (0.19–0.99) | 0.43 (0.19–0.995) | 0.43 (0.18–1.01) | 0.43 (0.18–1.009) | 0.43 (0.18–1.02) | 0.22 (0.01–3.87) |
| KWALI | 0.94 (0.44–2.01) | 0.95 (0.45–2.01) | 2.63 (0.23–30.30) | 0.84 (0.34–2.08) | 0.84 (0.34–2.08) | 0.25 (0.02–3.77) |
| COUNCIL AREA (HOME) | | | | | | |
| ABAJI | Reference | Reference | Reference | Reference | Reference | Reference |
| ABUJA MUNICIPAL AREA COUNCIL | 0.42 (0.20–0.86) | 0.42 (0.20–0.88) | 0.16 (0.01–2.23) | 1.22 (0.51–2.92) | 1.22 (0.51–2.92) | 1.64 (0.09–29.4) |
| BWARI | 0.77 (0.36–1.66) | 0.78 (0.36–1.68) | 0.42 (0.03–6.77) | 0.88 (0.35–2.22) | 0.88 (0.35–2.23) | 1.67 (0.08–32.60) |
| GWAGWALADA | 0.61 (0.32–1.18) | 0.61 (0.31–1.18) | 0.21 (0.02–2.52) | 0.74 (0.34–1.63) | 0.75 (0.34–1.64) | 0.61 (0.04–7.91) |
| KUJE | 0.56 (0.25–1.24) | 0.56 (0.25–1.25) | 0.94 (0.07–12.2) | 0.53 (0.22–1.28) | 0.53 (0.22–1.29) | 2.31 (0.12–43.29) |
| KWALI | 0.82 (0.37–1.79) | 0.82 (0.37–1.79) | 0.33 (0.03–1.79) | 1.10 (0.42–2.9) | 1.09 (0.41–2.89) | 4.05 (0.20–69.28) |
| OTHER†† | 0.52 (0.15–1.86) | 0.51 (0.14–1.84) | 0.20 (0.01–3.20) | 1.58 (0.30–8.18) | 1.60 (0.31–8.28) | 2.42 (0.11–54.06) |

*Adjusted for age and sex.

†Adjusted for age, sex, occupation, work council area, and history of immunocompromised state **"Other occupation" includes pharmacist, laboratory technician, non-physician clinician/paramedical professional. †† "Other home council area" represents any council area outside of the Federal Capital Territory.

BMI = body mass index.

protein IgG levels [8]. As a referral center, it is possible that healthcare workers at this tertiary care facility were exposed to more patients with COVID-19 or participated in diagnostics or treatment interventions that increased their risk of SARS-CoV-2 infection.

Results from the World Health Organization UNITY Studies Collaborative Group, which aims to synthesize data from seroprevalence studies across Africa (n = 151 studies overall; n = 7 studies from Nigeria), have shown substantial geographic and temporal variation [4]. SARS-CoV-2 seroprevalence estimates increased from 3.0% (95% CI: 1.0, 9.2%) in Q2 2020 to 65.1% (95% CI: 56.3, 73.0%) in Q3 2021 among all studies. During the current study period, seropositivity reported in regional or subnational Nigerian studies ranged from 9.3% (95% CI: 6.9, 11.6%) in Gombe State to 25.4% (95% CI: 19.3–32.3%) in Niger State [5]. In Nigeria's only national seroprevalence study conducted in Q3 2021, seropositivity was 78.9% (95% CI: 77.7, 80.0) [6]. No prior published community- or facility-based seroprevalence studies have been conducted in the Federal Capital Territory, which may partially explain the differences observed in the current study with prior research, as well as our inclusion of primary health care-level healthcare workers. Based on the increased risk for COVID-19 exposure that healthcare workers face, differences between general population and healthcare worker seroprevalence estimates are expected.

Seroprevalence reports from other parts of Africa have been limited, especially among healthcare workers, but the current study results appear plausible based on estimates from other reports during the pandemic. As of June 2020, IgG seroprevalence among Malawi tertiary healthcare workers was 12.3% (95% CI: CI 8.2, 16.5%) using a spike protein (S2 Fig) and nucleocapsid-based assay [19]. As of February 2021, IgG seroprevalence among urban tertiary care healthcare workers in Ethiopia ranged from 53.7% (95% Credible Interval (CrI): 44.8, 62.5%) in Addis Ababa to 56.1% (95% CrI: 51.1, 61.1%) in Jimma using a nucleocapsid-based ELISA assay [20]. As of March 2021, IgG seroprevalence among urban adults 35–59 years old in South Africa was 59% (95% CrI: 49, 68%) using a nucleocapsid-based ELISA assay [21]. These results mirror the temporal increase in seroprevalence observed in the current study.

The current study has several strengths, including recruiting and retaining a large cohort of frontline healthcare workers, including with close to 90% primary health care workers who provide highly accessible care. Previous seroprevalence research in Nigeria has focused on tertiary care healthcare workers and has not included longitudinal follow-up. The 6-month retention rate (93.5%) was high, strengthening the robustness of these results in a setting where longitudinal studies have been previously considered to be extremely difficult. Furthermore, this study used a validated dried blood spot-based assay for sample collection among participants that required minimal staff training, which increased the reach of data collection to settings where previous research has been limited.

This study also had some limitations, including some samples that were missing, mishandled, or inadequate for analysis, though this waned over time as study staff became more familiar with the blood spot sample collection technique. Further, contemporary community seroprevalence were not available nor evaluated, which is an important, additional route for SARS-CoV-2 transmission among healthcare workers. Also, while the assay has been validated in a US population, it has not been compared against pre-COVID-19 pandemic samples in Nigeria. Thus, it may be possible that there is some cross-reactivity with other antibodies, as has been observed in other settings in West Africa, such as Nigeria, Ghana, and Mali, leading to a higher measured seropositivity rate [22, 23]. However, the specificity of SARS-CoV-2 IgG ELISA assay is higher for spike protein-based rather than nucleocapsid-based assays, meaning the potential for cross-reactivity using the assay utilized here is relatively low. Further, the threshold used to define seropositivity in the current was based on optimization using receiver

operating characteristic curves, an approach that had 94.5% specificity in Mali [22]. This finding suggests that the observed seroprevalence rates in the current study are likely valid.

## Conclusions

This study demonstrated high baseline, 3-month and 6-month follow-up prevalence of IgG antibodies to SARS-CoV-2 during the first two waves of the COVID-19 pandemic in Nigeria using a dried blood spot-based assay among a cohort of frontline healthcare workers, including primary healthcare workers. A small proportion of participants had symptoms at baseline, and most were mild, which warrants further investigation. These results may have implications in state- and national-level disease pandemic modeling.

## Supporting information

**S1 Table. Baseline study participants' symptoms to SARS-CoV-2 by age group.**
(DOCX)

**S2 Table. COVID-19 symptoms, diagnostic testing, diagnosis, and hospitalizations among participants at baseline, 3-month, and 6- month follow-up.**
(DOCX)

**S1 Fig. Study flowchart.**
(TIF)

**S2 Fig. IgG levels of participants who were seropositive at baseline and corresponding month 3 and month 6 follow up levels.** *$P_{trend}$ = 0.08.
(TIF)

**S3 Fig. IgG levels of participants who were seronegative to SARS-CoV-2 at baseline and corresponding month 3 and month 6 follow up levels.** *$P_{trend}$ = < .0001.
(TIF)

**S4 Fig. Mixed model effect to evaluate association between baseline characteristics and IgG seropositivity of SARS-CoV-2.**
(TIF)

**S5 Fig. Association of IgG seropositivity to SARS-CoV-2 with participant exposure status at baseline.**
(TIF)

## Author Contributions

**Conceptualization:** Dike B. Ojji, Amelia Sancilio, Abigail S. Baldridge, Lisa R. Hirschhorn, Thomas McDade, Mark D. Huffman.

**Data curation:** Dike B. Ojji, Amelia Sancilio, Gabriel L. Shedul, Ikechukwu A. Orji, Joel Abu, Blessing Akor, Nana Ripiye, Funmi Akinlade, Douglas Okoye, Emmanuel Okpetu, Helen Eze, Emmanuel Odoh, Haruna Abubakar, Abubakar M. Jamda, Mark D. Huffman.

**Formal analysis:** Aashima Chopra, Abigail S. Baldridge, Lisa R. Hirschhorn, Thomas McDade, Mark D. Huffman.

**Funding acquisition:** Mark D. Huffman.

**Investigation:** Amelia Sancilio, Gabriel L. Shedul, Ikechukwu A. Orji, Joel Abu, Blessing Akor, Nana Ripiye, Funmi Akinlade, Douglas Okoye, Emmanuel Okpetu, Helen Eze, Emmanuel Odoh, Haruna Abubakar, Abubakar M. Jamda, Thomas McDade, Mark D. Huffman.

**Methodology:** Dike B. Ojji, Ikechukwu A. Orji, Aashima Chopra, Joel Abu, Funmi Akinlade, Emmanuel Okpetu, Abigail S. Baldridge, Lisa R. Hirschhorn, Thomas McDade, Mark D. Huffman.

**Project administration:** Dike B. Ojji, Gabriel L. Shedul, Ikechukwu A. Orji, Emmanuel Odoh, Abigail S. Baldridge, Priya Tripathi.

**Supervision:** Dike B. Ojji, Ikechukwu A. Orji, Joel Abu, Blessing Akor, Nana Ripiye, Douglas Okoye, Emmanuel Okpetu, Helen Eze, Emmanuel Odoh, Priya Tripathi, Haruna Abubakar, Lisa R. Hirschhorn, Mark D. Huffman.

**Validation:** Amelia Sancilio, Aashima Chopra, Priya Tripathi.

**Writing – original draft:** Dike B. Ojji, Mark D. Huffman.

**Writing – review & editing:** Dike B. Ojji, Amelia Sancilio, Gabriel L. Shedul, Ikechukwu A. Orji, Aashima Chopra, Joel Abu, Blessing Akor, Nana Ripiye, Funmi Akinlade, Douglas Okoye, Emmanuel Okpetu, Helen Eze, Emmanuel Odoh, Abigail S. Baldridge, Priya Tripathi, Haruna Abubakar, Abubakar M. Jamda, Lisa R. Hirschhorn, Thomas McDade.

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
