## [Decision Letter · Decision Letter 0]

8 Mar 2022

PGPH-D-21-01149

Nigeria Healthcare Worker SARS-CoV-2 Serology Study:  Results from a Prospective, Longitudinal Cohort

Dear Dr. Ojji,

Thank you for submitting your manuscript to PLOS Global Public Health. After careful consideration, we feel that it has merit but does not fully meet PLOS Global Public Health’s publication criteria as it currently stands. Therefore, we invite you to submit a revised version of the manuscript that addresses the points raised during the review process.

Please pay special attention to:

1. Elaborating more the baseline of the study:

a. methods - a dried blood spot IgG assay to determine the presence of SARS-CoV-2 specific IgG – not only refer to the literature, but describe

b. baseline seropositive individuals who reported COVID-19

c. how many reported COVID-19 in follow up (3- and 6 months)

d. COVID vaccination during study period

e. SARS-CoV-2 variants circulating during study period

2. Reformulating of the results and reorganize the graphical presentation of the results:

a. consider simplifying and/or combining figures and tables

b. explain age differences among developed symptomatic COVID compared to asymptomatic COVID

c. elaborate SARS-CoV-2 variants differences among developed symptomatic COVID compared to asymptomatic COVID

3. Rewriting the discussion - essentialy modify this section with a special attention to the recently published works on SARS-CoV-2 seroprevalence in Africa using the sources suggested by the reviewer

We invite you to make corrections to the manuscript, following the included below, detailed comments indicated by the Reviewers.

<ul><li> 

A rebuttal letter that responds to each point raised by the editor and reviewer(s). You should upload this letter as a separate file labeled 'Response to Reviewers'.<li> 

A marked-up copy of your manuscript that highlights changes made to the original version. You should upload this as a separate file labeled 'Revised Manuscript with Track Changes'.<li> 

We look forward to receiving your revised manuscript.

Kind regards,

Hanna Nalecz, Ph.D.

Academic Editor

Journal Requirements:

1. Please provide additional details regarding participant consent. In the ethics statement in the Methods and online submission information, please ensure that you have specified whether consent was written or verbal/oral. If consent was verbal/oral, please specify: a) whether the ethics committee approved the verbal/oral consent procedure, b) why written consent could not be obtained, and c) how verbal/oral consent was recorded. If your study included minors, please state whether you obtained consent from parents or guardians in these cases. If the need for consent was waived by the ethics committee, please include this information.

2. Please update the completed 'Competing Interests' statement. Please declare all competing interests beginning with the statement “I have read the journal's policy and the authors of this manuscript have the following competing interests:”.

Please remove the statement “The funders had no role in the design, conduct of the study; collection, management, analysis, and interpretation of the data; preparation, review, or approval of the manuscript; or decision to submit the manuscript for publication." in your 'Competing Interests' statement.

3. Please provide separate figure files in .tif or .eps format only and ensure that all files are under our size limit of 20MB.

4. In the online submission form, you indicated that your data will be submitted to a repository upon acceptance. We strongly recommend all authors deposit their data before acceptance, as the process can be lengthy and hold up publication timelines. Please note that, though access restrictions are acceptable now, your entire data will need to be made freely accessible if your manuscript is accepted for publication. This policy applies to all data except where public deposition would breach compliance with the protocol approved by your research ethics board. If you are unable to adhere to our open data policy, please kindly revise your statement to explain your reasoning and we will seek the editor's input on an exemption. Please be assured that, once you have provided your new statement, the assessment of your exemption will not hold up the peer review process.

Reviewers' comments:

Reviewer's Responses to Questions

**Comments to the Author**

1. Does this manuscript meet PLOS Global Public Health’s publication criteria? Is the manuscript technically sound, and do the data support the conclusions? The manuscript must describe methodologically and ethically rigorous research with conclusions that are appropriately drawn based on the data presented.

Reviewer #1: Partly

Reviewer #2: Yes

2. Has the statistical analysis been performed appropriately and rigorously?

Reviewer #1: I don't know

Reviewer #2: Yes

3. Have the authors made all data underlying the findings in their manuscript fully available (please refer to the Data Availability Statement at the start of the manuscript PDF file)?

Reviewer #1: Yes

Reviewer #2: No

4. Is the manuscript presented in an intelligible fashion and written in standard English?

Reviewer #1: Yes

Reviewer #2: Yes

5. Review Comments to the Author

Reviewer #1: The manuscript by Ojji et al highlights some very important data. I wish to make the following comments.

Methods

Serological testing: The authors have used a dried blood spot IgG assay to determine the presence of SARS-CoV-2 specific IgG. They refer the a published paper for methods. However, it would be important to have the following information regarding this dried spot assay in this manuscript itself, without just citing a previous published paper as this validity of this assay is a crucial part of this manuscript.

1. What was the sensitivity and specificity of this assay compared to standard neutralization assays or any commercial assays? How as it validated?

2. Does it detect IgG to previous coronavirus infections? Has this being validated in the Nigerian community?

3. Was this validated using seronegative blood samples prior to the COVID-19 pandemic? What was the result? On how many samples?

Results

1. How many of the baseline seropositive individuals reported COVID-19? Were all these infections asymptomatic?

2. In the population followed up at 3 months and 6 months, how many reported COVID-19? Of the individuals who seroconverted between 0 to 3 months and 3 to 6 month time points, how many seroconverted asymptomatically and how many were diagnosed to have COVID?

3. Did any individual receive any COVID vaccines at any of the time points?

4. What were the SARS-CoV-2 variants circulating during the time of the study? Was there any sequencing done in the country during the time of the study? Because the study was carried out from July 2020 to Jan 2021, there were no variants of concern circulating at that time?

5. Were there differences in ages in those who developed symptomatic COVID compared to asymptomatic COVID?

6. Based on the study results, the majority of infections in health care workers were asymptomatic? When delta variant was affecting the country, did Nigeria experience more symptomatic infection in HCWs?

Discussion

It would be important to discuss how these findings compare with studies done in a similar time period in health care workers in other countries.

Reviewer #2: In this paper, authors are providing a estimation of SARS-CoV-2 seroprevalence in a cohort of frontline, primary & tertiary care levels in Nigeria. The current study adds to current evidence as it looks not only at IgG seroprevalence cross-sectionally, but also provides insights into SARS-CoV-2 IgG Ab trends overtime (including sero-conversion and sero-reversion) with sampling repeated at M3 and M6 in the selected HCW cohort, with a high follow up rate (above (90%). Further, sampling was comprehensive, including allied workers and community health extension workers in addition to clinical staff. The data presented indicates that baseline IgG seropositivity (31%) was comparable to that reported previously in Nigeria and elsewhere (ie: DRC and Zimbabwe), but that it increased considerably over 6 months period (more than doubling to 70%). Of particular interest is the fact that this study was conducted prior to vaccines roll out in Nigeria and seem to corroborate existing evidence for widespread population exposure and higher infection rate than captured and reported by RT-PCR testing.

It is recommended that authors rewrite their discussion to reference recently published work on SARS-CoV-2 seroprevalence in Africa. The following would be of particular relevance :

• https://doi.org/10.1101/2022.02.14.22270934 (A systematic review and meta-analysis of standardised UNITY seroprevalence studies)

• https://doi.org/10.1093/heapol/czab133 (review of SARS-CoV-2 seroprevalence studies across 11 African countries)

While the authors thoroughly review their results, the abundance of figures and tables making the result section difficult to follow. Authors should consider streamlining and combining figures and tables to better highlight key findings from the study.

I recommend to accept the manuscript with major revisions

Major comments:

Line 101: authors indicate that “health care workers from 60 primary health care centers and University of Abuja Teaching Hospital” where identified. Yet on line 104 they state that multi-stage probability sampling was performed “among the 243 primary health care centers”. Please clarify this discrepancy.

Section on serological testing: authors should consider providing the specifications of the serological test used and its performance characteristics. Notably include existing validation data for testing on DBS vs serum. Have authors evaluated possible cross reactivity with other pathogens as reported in the region (doi: 10.3201/eid2701.203281) ?

Line 435: GISAID is a database dedicated to hosting genomics data. It is not clear how this would be relevant to the data from the current study. Authors may instead consider submitting data to Serotracker (https://serotracker.com/en/Data). Further, to comply with PLOS data policy, data and related metadata underlying reported findings should be deposited ahead of publication (https://journals.plos.org/globalpublichealth/s/data-availability).

Minor comments

Line 123: authors should consider providing study ethics approval nb and/or date

Line 124: REDCap was used to obtain informed consent: can authors clarify how that was done? was it an electronic signature ?

Line 128 to 131: content repeat line 95-97. Suggest deleting one of these instances to avoid redundancy.

Line 212 to 213: supplemental tables 3 and 4 are cited but do not seem to be included in the manuscript.

Line 279 to 293: Authors should consider updating these paragraphs to reflect currently available data about SARS-CoV-2 sero prevalence in Africa.

6. PLOS authors have the option to publish the peer review history of their article (what does this mean?). If published, this will include your full peer review and any attached files.

**Do you want your identity to be public for this peer review?** For information about this choice, including consent withdrawal, please see our Privacy Policy.

Reviewer #1: **Yes: **Neelika Malavige

Reviewer #2: No

---

## [Decision Letter · Decision Letter 1]

2 Oct 2022

PGPH-D-21-01149R1

Nigeria Healthcare Worker SARS-CoV-2 Serology Study:  Results from a Prospective, Longitudinal Cohort

Dear Dr. Ojji,

Thank you for submitting your manuscript to PLOS Global Public Health. After careful consideration, we feel that some minor revisions are still needed. Therefore, we invite you to submit a revised version of the manuscript that addresses the points raised during the review process.

We look forward to receiving your revised manuscript.

Kind regards,

Anete Trajman

Academic Editor

Journal Requirements:

1. Please provide additional details regarding participant consent. In the ethics statement in the Methods and online submission information, please ensure that you have specified whether consent was written or verbal/oral. If consent was verbal/oral, please specify: 1) whether the ethics committee approved the verbal/oral consent procedure, 2) why written consent could not be obtained, and 3) how verbal/oral consent was recorded. If your study included minors, please state whether you obtained consent from parents or guardians in these cases. If the need for consent was waived by the ethics committee, please include this information.

Additional Editor Comments (if provided):

This is a very well written manuscript on a relevant subject, many different results besides the prevalence itself are interesting, such as the absence of symptoms in most, indicating that infection was much more frequent than possibly reported by countries. A sentence on this would be nice in the discussion. While revising your text as recommended by reviewer 3, please also include a sentence on possible selection bias from the recruitment method (active response of an invitation, HCW with more concern would have possibly responded more positively). Congratulations on your work.

Reviewers' comments:

Reviewer's Responses to Questions

**Comments to the Author**

1. If the authors have adequately addressed your comments raised in a previous round of review and you feel that this manuscript is now acceptable for publication, you may indicate that here to bypass the “Comments to the Author” section, enter your conflict of interest statement in the “Confidential to Editor” section, and submit your "Accept" recommendation.

Reviewer #2: All comments have been addressed

Reviewer #3: All comments have been addressed

2. Does this manuscript meet PLOS Global Public Health’s publication criteria? Is the manuscript technically sound, and do the data support the conclusions? The manuscript must describe methodologically and ethically rigorous research with conclusions that are appropriately drawn based on the data presented.

Reviewer #2: (No Response)

Reviewer #3: Yes

3. Has the statistical analysis been performed appropriately and rigorously?

Reviewer #2: (No Response)

Reviewer #3: Yes

4. Have the authors made all data underlying the findings in their manuscript fully available (please refer to the Data Availability Statement at the start of the manuscript PDF file)?

Reviewer #2: (No Response)

Reviewer #3: Yes

5. Is the manuscript presented in an intelligible fashion and written in standard English?

Reviewer #2: (No Response)

Reviewer #3: Yes

6. Review Comments to the Author

Reviewer #2: (No Response)

Reviewer #3: In this study the authors evaluated baseline seroprevalence, rates of seroconversion (IgG- to IgG+) and seroreversion (IgG+ to IgG-), change in IgG concentration at 3- and 6-month follow-up, and factors associated with seropositivity. This data is important for the optimum management of public health in the study region & population. The authors must be commended for that.

My comments:

The authors have responded to the reviewers’ comments appropriately.

In table -2, it is unclear what models 1, 2 & 3 mean. It would be better to describe in methods section and to written below table-3.

7. PLOS authors have the option to publish the peer review history of their article (what does this mean?). If published, this will include your full peer review and any attached files.

**Do you want your identity to be public for this peer review?** For information about this choice, including consent withdrawal, please see our Privacy Policy.

Reviewer #2: No

Reviewer #3: **Yes: **Ramachandran Thiruvengadam

---

## [Editor Report · Decision Letter 2]

12 Dec 2022

Nigeria Healthcare Worker SARS-CoV-2 Serology Study:  Results from a Prospective, Longitudinal Cohort

PGPH-D-21-01149R2

Dear Dr Ojji,

We are pleased to inform you that your manuscript 'Nigeria Healthcare Worker SARS-CoV-2 Serology Study:  Results from a Prospective, Longitudinal Cohort' has been provisionally accepted for publication in PLOS Global Public Health.

Best regards,

Anete Trajman

Academic Editor

Thank you for your revised manuscript, and congratulations on your work.